# Neonates and Infants with Left Heart Obstruction and Borderline Left Ventricle Undergoing Biventricular Repair: What Do We Know about Long-Term Outcomes? A Critical Review

**DOI:** 10.3390/healthcare12030348

**Published:** 2024-01-30

**Authors:** Massimiliano Cantinotti, Vivek Jani, Shelby Kutty, Pietro Marchese, Eliana Franchi, Alessandra Pizzuto, Cecilia Viacava, Nadia Assanta, Giuseppe Santoro, Raffaele Giordano

**Affiliations:** 1Foundation G. Monasterio CNR-Regione Toscana, 56124 Pisa, Italy; cantinotti@ftgm.it (M.C.); pmarchese@ftgm.it (P.M.); eliana.franchi@ftgm.it (E.F.); apizzuto@ftgm.it (A.P.); cecilia.viacava@ftgm.it (C.V.); assanta@ftgm.it (N.A.); giuseppe.santoro@ftgm.it (G.S.); 2Helen B. Taussig Heart Center, Department of Pediatrics, Johns Hopkins Hospital, Baltimore, MD 21205, USA; vjani1@jhmi.edu (V.J.); shelby.kutty@gmail.com (S.K.); 3Adult and Pediatric Cardiac Surgery, Department of Advanced Biomedical Sciences, University of Naples “Federico II”, 80131 Naples, Italy

**Keywords:** congenital heart disease, pediatric, cardiac surgery

## Abstract

Background: The decision to perform biventricular repair (BVR) in neonates and infants presenting with either single or multiple left ventricle outflow obstructions (LVOTOs) and a borderline left ventricle (BLV) is subject to extensive discussion, and limited information is known regarding the long-term outcomes. As a result, the objective of this study is to critically assess and summarize the available data regarding the prognosis of neonates and infants with LVOTO and BLV who underwent BVR. Methods: In February 2023, we conducted a review study with three different medical search engines (the National Library of Medicine, Science Direct, and Cochrane Library) for Medical Subject Headings and free text terms including “congenital heart disease”, “outcome”, and “borderline left ventricle”. The search was refined by adding keywords for “Shone’s complex”, “complex LVOT obstruction”, “hypoplastic left heart syndrome/complex”, and “critical aortic stenosis”. Results: Out of a total of 51 studies, 15 studies were included in the final analysis. The authors utilized heterogeneous definitions to characterize BLV, resulting in considerable variation in inclusion criteria among studies. Three distinct categories of studies were identified, encompassing those specifically designed to evaluate BLV, those focused on Shone’s complex, and finally those on aortic stenosis. Despite the challenges associated with comparing data originating from slightly different cardiac defects and from different eras, our results indicate a favorable survival rate and clinical outcome following BVR. However, the incidence of reintervention remains high, and concerns persist regarding residual pulmonary hypertension, which has been inadequately investigated. Conclusions: The available data concerning neonates and infants with LVOTO and BLV who undergo BVR are inadequate and fragmented. Consequently, large-scale studies are necessary to fully ascertain the long-term outcome of these complex defects.

## 1. Background

The decision whether to perform biventricular repair (BVR) or univentricular palliation (UVP) in neonates and infants affected by multiple left heart obstructive lesions and borderline left ventricle (BLV) is always challenging [1,2,3,4,5,6,7,8,9,10,11,12,13,14,15,16], and knowledge regarding the long-term prognosis of these patients remains scarce. Various scores utilizing echocardiography data have been proposed as a means of predicting the efficacy of BVR in BLV [2,3,4,5,9,14]; however, these are solely based on short-term results [3,4,14], with a mean follow-up period of 5–6 years. Furthermore, these scoring systems are not without limitations, which include heterogeneity in the echocardiographic parameters and outcome measures evaluated, as well as the retrospective design during the score development [2,3,4,5,9,14]. Additionally, significant variations in inclusion criteria were observed among the studies from differing authors [2,3,4,5,9,14]. Moreover, the term “borderline left ventricle” encompasses a broad spectrum of complex cardiac defects characterized by one or multiple left-sided obstructions and diminutive left sections, with significant uncertainty regarding the optimal choice between BVR and UVP. The absence of a universally accepted definition of “borderline left ventricle” has resulted in the use of various definitions, such as critical left ventricular outflow (LVOT) stenosis, hypoplastic left heart complex, multiple left heart obstructive lesions, and small left heart structures. Additionally, neonates and infants with Shone’s complex [17,18,19,20,21,22,23,24,25,26,27] have typically been evaluated separately, even though these congenital heart defects (CHDs) are also characterized by multiple left-sided obstructions and borderline left structures. Despite the high volume of literature on surgical and percutaneous aortic stenosis (AS) valvuloplasty in pediatric patients [28,29,30,31,32,33,34,35,36,37,38,39,40,41,42,43,44,45,46,47,48,49,50,51,52,53,54], studies have not specifically focused on BLV despite neonatal AS being typically associated with a diminutive left ventricle (LV). Data regarding the long-term outcomes of neonates and infants with Shone’s complex [17,18,19,20,21,22,23,24,25,26,27] and critical neonatal AS [28,29,30,31,32,33,34,35,36,37,38,39,40,41,42,43,44,45,46,47,48,49,50,51,52,53] undergoing BVR are also limited. The aim of the present study is to systematically review data pertaining to the long-term outcomes of neonates and infants with one or multiple LVOT obstructions and BLV who have undergone a biventricular correction. 

## 2. Methods

In February 2023, we conducted a review study within 3 different medical search engines (the National Library of Medicine, Science Direct, and Cochrane Library) for Medical Subject Headings and free text terms including “congenital heart disease”, “outcome”, and “borderline left ventricle”. The search was refined by adding keywords for “Shone’s complex”, “complex LVOT obstruction”, “hypoplastic left heart syndrome/complex”, and “critical aortic stenosis”.

The titles and abstracts of articles identified by this strategy were evaluated and excluded if (i) the reports were written in languages other than English (1 study); (ii) studies did not report a data follow-up of at least 3 years (13 studies); (iii) there were duplicate data (6 works); (iv) the reports mixed neonates and infants with older children (9 studies); (v) the reports evaluated only single lesions (e.g., isolated aortic stenosis, excluding multiple stenosis and borderline LV) (7 studies) (Figure 1).

## 3. Results

Of 51 studies, 36 were excluded for the above-mentioned criteria, leaving 15 studies for the final analysis. Three broad categories of studies were identified: those specifically designed to evaluate neonates and infants with BLV [1,3,4,6,7,10], those on Shone’s complex [22,23,24], and those on aortic stenosis [28,30,31,43,47]. 

### 3.1. CHD Definition and Groups

Various definitions have been utilized to describe LV obstruction with borderline LV, including BLV [1] (Table 1), critical left ventricular outflow (LVOT) stenosis [3,4], hypoplastic left ventricle with mitral stenosis [6], hypoplastic left heart complex [8,11], multiple left heart obstructive lesions [7,10], and small left heart structures [9]. Two main categories of BLV definitions may be identified: one includes in BLV a series of defects characterized by two or more diminutive left-sided lesions (e.g., mitral valve-MV, aortic valve AoV, LVOT, LV) [1,6,7,10,11,55], while the other is focused on the presence of a critical LVOT obstruction [3,4,8] (Table 1). While LV hypoplasia was a criterion of inclusion for most authors [1,3,6,7,8,10,11], not all authors used this inclusion criterion [4,9]. Notably, one study had the presence of aortic coarctation [9] as the primary inclusion criterion, while others excluded patients with isolated aortic arch obstruction [1]. Z-scores were employed to determine the degree of mitral and aortic valve hypoplasia, but different Z-score sources were employed by various authors [56,57,58], and at times the nomograms employed were not indicated at all [6,10,11]. Established exclusion criteria were discordant atrio-ventricular and ventricular arterial connections [1,3,4,7,9], aortic or mitral atresia [1,3,4,6,8,10], atrioventricular septal defects [1,6,7,10], or other complex associated CHDs [3,7,8,10].

The definition of Shone’s complex is also not consistent throughout the available literature. It has been defined as mitral stenosis plus at least one [19,22,24] or two [23] LVOT obstructions. Furthermore, the literature on aortic stenosis is also greatly heterogenous [54], with articles focused on the results of surgical valvuloplasty [28,30,31] or percutaneous valvuloplasty [43,47]. Finally, there is a challenge obtaining data on children with BLV since they have often been excluded [5,28,30,32,49,52,53] or grouped together with older children [31,41]. Several studies used fixed cut-off values to exclude patients from BVR [5,28,30,49], such as aortic valve annulus < 5 mm [28,30] or <4.5 mm [5], MV annulus < 7 mm, LV end-diastolic volume < 20 mL/m^2^ [30], and LV long axis < 80% of the LV [49]. Other studies utilized more generic exclusion criteria, such as borderline LV [38,53] or multiple LVOT obstruction [32,52]. The present study focuses on three articles on surgical valvuloplasty performed in neonates and infants with critical AS [28,30,31] and two articles on balloon valvuloplasty [43,47].

### 3.2. Follow-Up Duration

Our investigation revealed the finding of very limited data on long-term follow-up of BLV (Table 2), with only seven studies [1,3,4,6,7,8,10] reporting follow-up data of at least 5 years. However, not all the studies [7,8,9,10] completed the 5-year follow-up for all subjects [1,7,8,9,10]. The longest follow-up interval was, on average, 8 years [1], but once again, not all the subjects completed the follow-up. Follow-up intervals ranged from 0.1 to 16.4 years [1]. Some studies had a very limited sample size (e.g., <20 subjects) [6,8,11], while others had moderate to large sample sizes (ranging from 39 to 71 subjects) [1,7,9,10] or very large sample sizes (e.g., >100 subjects) [3,4]. All the studies, except for two [3,4], had a retrospective, single-center design. 

In the studies pertaining to the Shone complex, only three retrospective studies conducted at single centers [22,23,24] with sample sizes ranging from 27 [23] to 121 subjects [22] reported follow-up data for at least 5 years. The duration of follow-up was longer compared to studies pertaining to BLV, varying from 4.4 ± 4 years [23] up to 7.9 years [24].

The included studies on aortic stenosis were all single-center with retrospective design [28,30,31,43,47]. Here, mean sample sizes varied from 37 [30] to 84 subjects [28], while follow-up varied from a mean of 3.2 years [43] to >8 years [31], with a total duration of follow-up of 17.7 years in some cases [28].

### 3.3. Survival/Transplant-Free Survival for BVR

In the context of BLV, the definition of successful BVR varies among authors. Some describe it as “survival alone” [3,4,7,8], while others use “transplant-free survival” [1,9,10]. The reported rates of successful BVR ranged from 70% [3] to 82% [1] at 5 years. There are only a couple of studies [1,6] reporting data on survival at 10 years of follow-up, with a survival rate ranging from 77% [1] (transplant-free survival) to 88.5% (survival alone) [6]. Studies evaluating Shone’s complex reported transplant-free survival rates ranging from 86% [22] to 61.3% [24] at 10 years. In studies on AS [28,30,31,43,47], different time intervals have been evaluated to estimate survival rate and freedom from reintervention. Survival at 10 years varied from 42% [47] to 86% [43], and at 15 years the survival rates varied from 85% [28] to 64% [31]. Late deaths were rare, with survival rates being very similar at 1 year and 10 years [30,43].

### 3.4. Freedom from Reintervention

Reports on reintervention rates in borderline LV studies have been inconsistent or not reported at all [1]. Even when reported, obtaining a real estimate is difficult due to very limited [7,9,11] or varying [8] follow-up duration. Freedom from reintervention at 1 year varied from 50% [7] to 61% [9], while at 3 years, it ranged from 25% [11] to 50% [5]. Multiple interventions were frequently required [4,5,6,10], with a series [10] of 72 hypoplastic left heart complexes with a mean follow-up of 5.9 years (range 2.0 to 12.1 years) requiring an estimated 1.9 interventions per patient [10]. In the Cardiac Heart Surgery Society (CHSS) series, 19.2% required two reinterventions and 5.7% required three reinterventions [5]. Studies pertaining to Shone’s complex evaluated freedom from reintervention on longer-term follow-up, with freedom from reintervention estimates varying from 72% [22] to 61.3% [24] at 10 years. In all the studies, reinterventions on the mitral valve (repair or replacement) were the most common, along with subaortic membrane resection or more invasive interventions on the LVOT such as Konno or Ross operations [22,23,24]. 

Finally, in aortic stenosis freedom from reintervention varied from 21% [47] to 65% [28] at 10 years and from 18% [28] to 64% [31] at 15 years. The limited available data make it difficult to compare surgical and percutaneous valvuloplasty. However, a recent review and meta-analysis [54] demonstrated no significant difference in survival between surgical and percutaneous valvuloplasty for congenital aortic stenosis, although the incidence of reintervention was higher in percutaneous valvuloplasty (*p* < 0.001).

### 3.5. Z-Scores Increase at Follow-Up

All the studies demonstrated a significant growth of left ventricular structures and their Z- scores after BVR. Left ventricle end-diastolic diameter Z-scores increased significantly [6,8,11,22] (*p* from <0.02 to <0.001), as well as LV mass (*p* from 0.11 to <0.005) [6,7] and LV/RV ratio (*p* < 0.01) [11]. Aortic annulus Z-scores (*p* from <0.05 to <0.001) [7,9,22] and MV Z-scores (*p* all <0.001) also increased [8,9,11,19,22,23]. 

### 3.6. Clinical Outcome, Incidence of Pulmonary Hypertension, Need of Medications

Limited data are available on New York Heart Association (NYHA) class and the need for medication during follow-up. However, in the available studies, most patients were in class NYHA I at the last follow-up examination [6,7,10]. In a small series of eight children with hypoplastic left ventricle and mitral stenosis with a mean follow up of 6.5 years (±4.5 years), most cases were in class NYHA I or II, with only three patients requiring medication. Another study [7] of over 39 hypoplastic left heart complexes, with a mean follow-up f of 34 months (range 177 months), reported that 74% of patients were in class NYHA I, while the remaining 26% were in class NYHA II, and only 19% of all patients required medications. 

A study on a series of 43 patients with Shone’s complex demonstrated that at 10 years, 82.6% of survivors were in NYHA I, with mild or less mitral regurgitation present in 66.7%, normal LV function was observed in 79.2%, and 91.7% were free of any LVOTO [24]. Similarly, another series reporting on 27 Shone cases found that at 15 years, all survivors were either NYHA class I or II, 30% had moderate LV dysfunction, and 70% were free of residual obstruction [23].

The occurrence of pulmonary hypertension has been infrequently reported in the literature [1,6,7]. In the included studies, the incidence varies from 7.1% [1] (24 hypoplastic left heart with a mean follow-up of 8 years) to 44.1% [22] (122 Shone’s complex patients with a mean follow-up of 7.2 years), and even as high as 57.1% [6] (8 hypoplastic left ventricle+ mitral stenosis with a mean follow-up of 6.5 years).

### 3.7. Risk Factors for Poor Outcome and Reintervention

Several echocardiographic parameters have been evaluated to determine which ones are more predictive of poor outcomes following BVR. Studies have demonstrated that a lower aortic annulus Z-score [28,43,47], MV annulus Z-score [43,47], LV end-diastolic Z-scores [6,31,47], and LV dysfunction [28,47] were all predictors for successful BVR. Both the presence and the degree of endocardial fibroelastosis (EFE) were risk factors for poor outcome on univariate analysis [3,4,28,31]. Moderate or severe EFE was either the only factor that remained significant on multivariate analysis [43] or the combination of moderate or severe EFE with either lower aortic valve Z-scores and younger age [3] or LVOT diameters < 4 mm [4]. EFE has been found to be a predictor of death, even in cases of adequate LV function [4]. Furthermore, EFE has been shown to be a predictor for reintervention [28] along with ventricular dysfunction [28], the presence of multilevel stenosis [31], a small aortic annulus Z-score [4,43], a lower aortic root, and LV end-diastolic diameters [43]. In a study of 72 hypoplastic left heart complexes [10], a moderate to large ventricular septal defect (odds ratio-OR- = 0.22, *p* = 0.001) was found to be the strongest predictor of BVR failure, followed by a unicommisural valve (OR = 16, *p* = 0.006) and mitral valve Z-score (OR = 2.2, *p* = 0.002).

## 4. Discussion

Our critical assessment of the literature outlined how comprehensive data on the long-term outcome of neonates and infants with BLV undergoing BVR are limited and fragmented [2,3,4,5,9,10,11,12,13,14,17,18,19,20,21,22,23,24,25,28,29,30,31,32,33,34,35,36,37,38,39,40,41,42,43,44,45,46,47,48,49,50,51,52,53]. Although many studies have focused on medium/short-term results (e.g., 5 years survival) [1,3,4,6,7,8,9,10,11], limited knowledge exists on the long-term prognosis of these critical patients, not only regarding survival but also regarding the incidence of reintervention, complications, or quality of life. Additionally, the lack of standardization of disease definition posed a challenge to both the search and the comparison of data, leading to significant heterogeneity in the inclusion criteria. The term “borderline left ventricle” is generally employed to describe a series of CHDs characterized by one or multiple inflow and/or outflow left-sided stenosis and/or small left heart structures [1,3,4,6,7,8,9,10,11] including critical aortic stenosis [3,4], aortic coarctation (with or without aortic arch hypoplasia) [9], and Shone’s complex [22,23,24]. Different studies, however, have been focused on specific conditions including aortic stenosis and/or LVOT obstruction [3], or aortic coarctation [7], as the main criteria of inclusion. Shone’s complex has been generally considered separately [22,23,24], despite belonging to the same spectrum of disease. Furthermore, even the definition of Shone’s complex is not completely standardized yet. Shone’s complex was originally described as a parachute mitral valve, supra-annular mitral ring, subaortic stenosis, and coarctation of the aorta [22,23,24]. Most series, however, have defined the complex as any mitral valve disease of varying categories plus at least one additional level of left heart obstruction [22,23,24]. While there is a vast literature [54] on aortic stenosis in the pediatric age, even encompassing critical aortic stenosis, it remains a challenge to derive data specifically on children with borderline LV, since they have often been excluded [5,28,30,32,49,52,53]. 

Discrepancies in the inclusion criteria among different studies hamper the possibility to compare data and to perform a meta-analysis. Furthermore, it is difficult to compare data originating from different eras, with time periods from the late 1990s [3,4,10,11] to the present day [1] accompanied by their different outcomes due to the improvement of diagnostic interventional strategies. Even Z-scores employed for echocardiographic diagnosis and disease severity estimation employed by older works [56,57,58] presented significant limitations that have been overcome by more recent works [59,60,61]. Older Z-scores [58] tend to overestimate the disease severity [59]. Thus, authors who utilized older nomograms [3,7,10] probably included even milder forms of LV obstructive diseases in the definition of “borderline left ventricle”, which may have contributed to a more favorable outcome.

Transplant-free survival rates are acceptable at 5 years, ranging from 70% [3] to 82% [1]. Despite the limited data, transplant-free survival at 10 years also shows promising results, ranging from 61.3% [24] to 88.5% [6], with a survival reduction between 5 to 10 years follow-up of around 4–5% [1,19,22,23]. Recent studies [1,6,8,22] have shown slightly better outcomes than older series [10,11,31], suggesting that progress in surgical techniques may have contributed to improved outcomes. Similarly, it is also true that progress has been made regarding surgical techniques for univentricular correction, and consequently lifetime expectancy has increased [26]. A recent multicenter Australian registry [26] with over 301 lateral tunnel Fontan procedures performed between 1980 and 2014 demonstrated that overall survival at 15 and 25 years was 90% (95% confidence interval [CI]: 86–93%) and 80% (95% CI: 69–91%), respectively. Although biventricular Fontan procedures (such as those performed in borderline LV) have usually demonstrated more favorable results, no significant difference emerged in this study [26] among the outcomes of single- or biventricular total cavo-pulmonary connection. 

Neonates and infants with borderline LV who have undergone BVR continue to face a significant burden of reintervention. One or multiple reinterventions are often required [6,7,8,9,10,11], with up to 50% [7] or 60% [8,9] of patients requiring it within the first year of life. Early reintervention may indicate inappropriate decision making and is often associated with poor outcomes [5]. In fact, a short interval after the first reintervention, particularly within the first 30 days, has been identified [5] as a risk factor for mortality. As stated previously, multiple reinterventions are often necessary [5,10] and the available data indicate that freedom from reintervention in long-term follow-up is limited to 20% of cases [47] and up to 18% at 15 years [28]. 

Several echocardiographic markers have been identified to predict BVR failure. These markers include lower aortic and mitral valve annulus [28,43,47], LV end-diastolic volume [6,31,47], LV dysfunction [28,47], and endocardial fibroelastosis [3,4,28,31,43]. Endocardial fibroelastosis has been found to be a strong predictor of death [4,43] and the need for reintervention [28]. Despite limited evidence, the available data suggest that most patients who underwent BVR for BVL are in good clinical status [6,7,10,23,24]. Furthermore, long-term medications are typically only required in a minority of cases.

Incidence of pulmonary hypertension, which is a worrisome complication even in successful BVR without significant residual stenosis [34], was rarely described [1,6,7], varying from 44.1% [22] up to 57.1% [6] at medium-term follow-up. Despite the normalization of Z-scores of left sections that has been described by multiple authors [6,7,8,9,11,22], even in the lack of significant residual stenosis a moderate LV systolic [23,24] and diastolic dysfunction may persist and contribute to the maintenance or development of pulmonary hypertension. Unfortunately, however, little to no investigation has been conducted on LV diastolic function. 

### Strength and Limitations

This paper has several strengths. This is one of the first studies that has tried to evaluate together pathologies that have usually been considered separately (hypoplastic left heart complexes, Shone’s anomalies, critical aortic stenosis) despite belonging to the same spectrum of the disease (e.g., one or multiple LVOT obstruction and BLV). This is also one of the few papers that has tried to address the long-term outcome of BLV, including some aspects that have been poorly evaluated so far (e.g., the clinical status, the presence of pulmonary hypertension).

This paper also has some limitations, including the lack of homogenous definitions of the same disease and the fact that the choice of different end points hampered the possibility to perform a metanalysis of the current data. Even comparisons among such different data (and coming from different eras) resulted in difficulties. Large, multicenter studies with clear and uniform disease definitions and well-defined outcomes are advised for a better understanding of the long-term outcomes of complex CHDs characterized by one or multiple LVOT obstruction and BLV. 

## 5. Conclusions

Data on medium- and long-term outcomes of neonates and infants with multiple LVOT obstruction and borderline LV undergoing BVR are limited and fragmented. While survival rates have improved, patients who underwent BVR remain burdened by high reintervention rates. It is necessary to conduct large studies with standardized criteria for inclusion and exclusion, outcomes, and evaluation of clinical and echocardiographic variables, as well as with longer and more uniform follow-up intervals, to gain a better understanding of the long-term outcomes in this specific patient population. Our critical appraisal of the available literature may help guide clinicians in both parental counseling and clinical decision making, while also laying the foundation for future standardized studies regarding BVR for neonates and infants with BLV and multiple LVOT obstructions.

## Figures and Tables

**Figure 1 healthcare-12-00348-f001:**
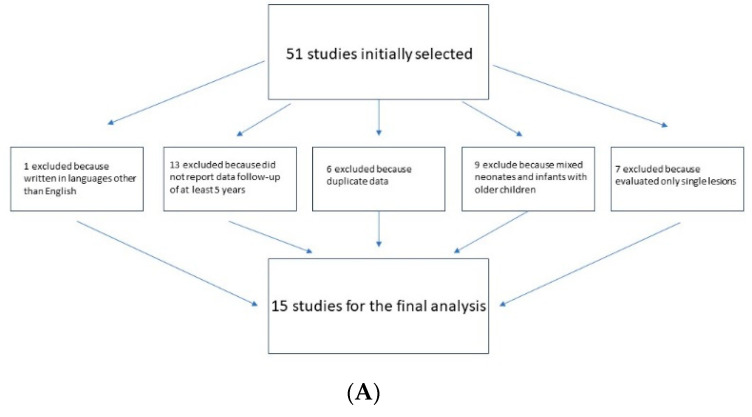
(**A**) Diagraph of inclusion/exclusion criteria. (**B**) Flow diagram for new review.

**Table 1 healthcare-12-00348-t001:** (**A**) Borderline LV definition according to different authors. (**B**) Shone’s complex definition according to major studies.

**(A)**		
**Authors**	**Inclusion Criteria**	**Exclusion Criteria**
*Two or more left-sided diminutive lesions*		
Kang SL, 2022, Toronto [1]	b-HLH two or more: − LVEDVi < 20 mL% by echo − Non apex forming LV − EFE − AoV or MV z score < 2 − Valvular or sub valvular LVOT − MV obstruction Z-score source: none indicated	− Only Ao or MV Z-score < 2 with associated obstruction − Valvular atresia − Discordant AV or VA connections − AVSD − Isolated Ao Arch obstruction
Cavigelli Brunner A, 2012, Zurigh [7]	<3 months, with patent Ao and MV two or more (1) MS, MV hypoplasia, or parachute MV (2) LVOT: stenosis diameter < than normal Ao annulus (3) AS, AoV hypoplasia (4) Ao arch: hypoplasia, isthmic hypoplasia, coarctation, or interrupted Ao arch type A (5) LV: LV/RV long-axis ratio <0.8 or small LVEDV Z-score source: Daubeney et al. [58]	Non concordant AV or VA connection, TAPR, or AVSD
Swartz ML, 2001; Boston, USA [10]	two or more (1) MS, MV hypoplasia, or parachute MV (2) LVOT: stenosis, diameter < than normal Ao annulus (3) AS or hypoplasia (4) Ao arch: hypoplasia, isthmic hypoplasia, coarctation, or interrupted Ao arch type A (5) LV: LV/RV long-axis ratio <0.8 or small LVEDV Z-score source: Daubeney et al. [58]	MA, AA, TAPVR, AVSD, TGA, truncus arteriosus, DORV, or interrupted Aortic arch type B
Tchervenkov CI, 1998, Montreal, Canada [11]	Multiple Hypoplastic structures of the left heart-aorta complex including MV, LV, LVOT, and the aortic valve. Anterograde flow through the Ao Z-score source: none indicated	NR
Mart CR et al., 2014, USA [55]	HLCH − MV, AV z score < 2 − LV hypoplasia without EFE − Hypoplasia of the LVOT, Asc Ao, and aortic arch with or without Aco − Holosystolic antegrade flow in the Asc Ao − Ductal dependency with bidirectional ductal flow Z-score source: Sluysman et al. [57]	AA, MA
Shimada M, 2019, Osaka, Japan [6]	HLV defined as having a z score < 2 and Hypoplastic or dysplastic MV Z-score source: none indicated	AA, MA, interrupt Ao arch. AVSD
*Critical left ventricle outflow obstruction*		
Lofland GK, 2001, CHSS, USA [3]	Critical LVOTO: − Moderately or severely reduced LV − Systemic perfusion dependent on RV output via a PDA Z-score source: Daubeney et al. [58]	− Large VSD or − associated cardiac anomalies judged to be of worse prognostic significance than their LVOTO − Abnormal AV or VA connections − AA
Hickey EJ, 2007, CHSS, USA [4]	Critical neonatal LVOTO: stenosis occurring at any level from the subvalvular region to the innominate artery with or without LV hypoplasia, such that the systemic circulation was ductus-dependent Z-score source: none indicated	AA, MA, interrupted Ao arch, Abnormal AV, or VA connections
Freund 2015, Utrecht, Netherlands [8]	Critical AS with hypoplasia of the LV, congenital MS with comparable HLV ° Z-score source: Pettersen et al. [57]	AA, MS, AS and MA, AS, DORV, TA, DILV
*Aortic coarctation plus diminutive left sections*		
Plymale JM, 2017, Milwaukee, WI, USA, [9]	Infants ≤2 months undergoing aortic arch repair with AoV and/or MV hypoplasia, (Z-score ≤ −2) Z-score source: Pettersen et al. [57]	Abnormal AV or VA connections
**(B)**		
** Authors **	** Inclusion Criteria **	** Exclusion Criteria **
*MS and at least one more defect*		
Nicholson GT, 2016, Atlanta and Houston, USA, [22]	ShoneMS+ at least one Subvalvular AS AS Supravalvular AS Aortic arch hypoplasia Aco	MS and AS palliated for UVP
Malhotra SP, 2008, Colorado, USA [24]	MS and one or more LVOT MS (Hammock or arcade MV, n = 9; parachute n = 12; supramitral ring n = 11) LVTO (subaortic stenosis, n = 25; aortic stenosis, n = 24; hypoplastic arch, n = 20; coarctation, n = 38) 213	1110, Patients undergoing UVP
*MS and at least two or more defects*		
Brown JW, 2005, Indianapolis, USA [23]	Shone MS+ at least two other left heart obstructive lesions (subaortic stenosis, n = 16; valvular aortic stenosis, n = 7; bicuspid aortic valve, n = 24; aortic coarctation n = 18)	Significant LV hypoplasia not suitable for BVR
Delmo Walter EM, 2013, Berlin [19]	Any type of MS+ LVOTO ° Aco and/or arch hypoplasia °	Patients undergoing UVP, straddling of TV or MV, uAVSD, multiple VSDs, unroofed coronary, sinus, DORV, and non-apex LV

AA = aortic atresia, Aco = aortic coarctation; Ao = aorta; AS = aortic stenosis, AoV = aortic valve, AV = atrio-ventricular, AVSD = atrioventricular septal defect, DILV = double inlet left ventricle, DORV = double outlet left ventricle, HLC = hypoplastic left heart complex; HLV = hypoplastic left ventricle; EFE = endocardial fibo-elastosis, MA = mitral atresia, MS = mitral stenosis, MV = mitral valve, LV = left ventricle, LVOT = left ventricular outflow tract, LVOTO = left ventricle outflow obstruction; LVEDD = left ventricle end-diastolic diameter, LVEDV = left ventricle end-diastolic volume, LVEDVi = left ventricle end-diastolic volume indexed, NR = not reported, PDA = patent arterial duct, RV = right ventricle, TA = tricuspid atresia, TAPR = total anomalous pulmonary venous return; TGA = transposition of the great arteries, VA = ventricular-arterial; ° definition of Tchervenkov CI; BVR = biventricular repair, MS = mitral stenosis, TV = tricuspid valve, uAVSD = unbalanced atrioventricular septal defect, UVP = univentricular palliation, VSD = ventricular septal defect.

**Table 2 healthcare-12-00348-t002:** (**A**) Major studies on outcomes in borderline LV attempting BVR. (**B**) Major studies on outcomes in Shone undergoing BVR. (**C**) Major studies on outcomes in neonates and infants with aortic stenosis undergoing either surgical/percutaneous valvuloplasty.

**(A)**					
**Authors**	**Sample Size**	**F-Up Time**		**Survival**	**Freedom from Redu**
* **Larger works (e.g., >50 subjects)** *					
Hickey EJ, 2007, CHSS, USA [4]	223 UVP, 139 BVR (1994–2001)	5 yrs		5 yrs UVP 62% ± 3% BVR 71% ± 4%	3 years 64% 19.4% two redu 5.7% three redu
Lofland GK, 2001, CHSS, USA [3]	116 BVR, 179 UVP (1994–2000)	5 yrs		5 yrs BVR 70% UVP 60%	NR
Schwartz ML, 2001, Boston, USA [10]	72 HLHC (1988–1997)	5.9 (2.0–12.1) yrs		BVR success 81% *	1.9 (range 0 to 7) per patient interventions
Kang SL, 2022, Toronto [1]	54 bHLH with BVR (2003–2015)	8.0 (0.1–16.4) yrs		TX free survival 1 yrs 96% 5 yrs 82% 10 yrs 77%	NR
* **Smaller works (e.g., <50 subjects)** *					
Cavigelli-Brunner A, 2012 Zurigh [7]	39 HLHC (1.30 days) (1990–2006)	34 (1–177) months		87% 74% NYHA I 26% NYHA II	1 years 50% 34 months 38%
Tchervenkov CT, 1998, Montreal, Canada [11]	11 HLHC + Aco with hypoplastic Ao arch (1988–1998)	44+/35 mths		8 yrs 63%	3 yrs 25%
Shimada M, 2019, Osaka, Japan [6]	8 HLV + MS (2001–2014)	6.5 (±4.5) yrs		10 yrs 88.5% All NYHA I or II	Redu in two (one multiple intervention)
**(B)**					
** Authors **	** Population **	** First Surgery **	** F-Up **	** Survival **	** Freedom from Redu **
Nicholson GT, 2016, Atlanta and Houston, USA, [22]	121 MS and 1 or more LVOTO (1978–2010) 28 days (0 days–17.3 years) MS + ACO 76.9% SubAS 42.1 AS 43% SupraAS (5%)	− 75 ACo − 22 MVP − 23 SubAo res	7.2 (0.1–35.5) yrs	Tx free survival 10 yrs 86% (14 of 17 deaths within first 10 mths of life)	10 yrs 71.7% Mean of two surgical/percutaneous intervention per patient
Malhotra SP, 2008, Colorado, USA [24]	43 MS and 1 or more LVOTO (1987–2007) Stage repair 30 6.5 days (1 day-1.6 yrs) TX 13 patients 22 (2–76) days	− 24 Aco − 2 Sub-AS − 1 RK and MVP, − 1 RK and MVR, − 1 MV ring, − 1 SAV	7.9 yrs	Survival staged repair 1 yr 96.2% 5 yrs 88.0%, 10 yrs 83.1% 82.6% in NYHA I	on LVOT 60.6% on MV 83.7% MVR 5 yrs 81.4% 10 yrs 73.2%
Brown JW, 2005, Indianapolis, USA [23]	27 MS and 2 LVOTO (1978–2003) 9.0 months (2 days-3 yrs) MS+ AS 26% Aco 67% SubAS 59%	15 Aco 4 VSD closure 3 MV ring 2 MVP 2 Sub-Ao res. 2 BAV	4.4 ± 4.4 yrs	TX free survival 5 yrs 96% 10 yrs 93% 15 yrs 89% All NYHA class I or II	82%
**(C)**					
**Authors**	**Population**	**F-Up**		**Survival**	**Freedom from Redu**
* **Surgical valvuloplasty** *					
Galoin-Bertal C, 2016, France [28] Period: 1994–2012	84 critical AS Age < 4 months	4.2 yrs (1 day–17.7 yrs)		5 yrs 87% 15 yrs 85%	5 yrs 51% 10 yrs 65% 15 yrs 18%
Brown JB, 2006, USA [31] Period: 1978–2000	66 critical AS SAV Age 15.1+/19.6 days	8.2 ± 6.2 yrs		15 yrs 94% in isolated AS 64% in AS + CHD All NYHA I or II	Freedom from AoV reoperation 5 yrs 83% 15yrs 64% 20 yrs 60%
Hawkins JA, 1998, Salt Lake, USA [30] Period: 1986–1996	37 critical AS SAV Age: 26 ± 21 days	Up to 11 yrs		1 yr 78% 10 yrs 73.4%	1 yrs 73% 10 yrs 55%
* **Percutaneous valvuloplasty** *					
Han RK, 2007, Toronto [43] Period: 1994–2004	53 Neonatal AS BAV Age 3.5 days (range 1 to 30 days §)	3.2 yrs (5 days–10.9 yrs)		1 year 86% 10 yrs 86%	1 year 68% 5 yrs 56% 10 yrs 33%
Latiff HA, 2003, Australia [47] Period: 1988–1998	42 As Age < 6 months 19 (1–180) days 0–7 days 16 8–30 days10 1–6 months 16	53 months (6 mths–9 yrs)		10 yrs All 72% 0–7 days 42% 8–30 days 65%, 1–6 months 93%	5 yrs 70% 10 yrs 21%

Ao Arch = aortic arch, AoV = aortic valve, AVr = aortic valve replacement, BAV = balloon aortic valvotomy, bHLH = borderline hypoplastic left heart, CHSS = cardiac heart surgeons’ society, HLHC = hypoplastic left heart complex, HLV = hypoplastic left ventricle, Int CC = interventional cardiac catheterization, BVR = biventricular repair, NYHA = New York Heart Association, MV = mitral valve, Pa = pulmonary artery, PH = pulmonary hypertension, SAV = surgical aortic valvotomy, surg = surgery, TX = transplant, UVP = univentricular palliation; f-up = follow-up. * Failure of biventricular repair was defined as takedown to a univentricular repair, cardiac transplantation, and/or death. AS = aortic stenosis, Aco = aortic coarctation, LV = left ventricle, LVOTO = left ventricle outflow tract obstruction, MS = mitral stenosis, MVP = mitral valve plasty, MVR = mitral valve replacement, SR = staged repair, SubAo res = subaortic resection, SubAS = sub-aortic stenosis, Supra, As = supra-aortic stenosis, RK = Ross Konno, VSD = ventricular septal defect. Ao = aorta, CHD = congenital heart disease, § Median CHSS score o (−46 to 55) 53% negative CHHS score 47% positive CHSS score.

## Data Availability

Not applicable.

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
