# Peer review of "Neonates and Infants with Left Heart Obstruction and Borderline Left Ventricle Undergoing Biventricular Repair: What Do We Know about Long-Term Outcomes? A Critical Review"

_healthcare, 2024, doi:10.3390/healthcare12030348_

Round 1

Reviewer 1 Report

Comments and Suggestions for Authors

Dear authors, 

I suggest the following to improve the manuscript:

1. Have the manuscript revised by a native English speaker familiar with the subject. 

2. Did you plan on doing a systematic review? If yes, please report if you have registered it or not? Please report it in the form of a systematic review. If not, why didn't you do it as a systematic review? 

3. In your exclusion criteria, you state that studies were excluded if they did not report at least 5 years of follow-up, but in your tables and results you also talk about studies with follow up of less than 5 years. Please explain. 

4. In line 264 you state "MV Z-score (OR=2.2, P=0.002, with an odds ratio of 1 for 1 unit decrease in Z-score).", what do you mean by odds ratio of 1 for 1 unit decrease? 

5. Describe what the AoV and mitral valve z score is. 

Comments on the Quality of English Language

The English language needs improvement. 

Author Response

Revisor 1

Many thanks for your revision:

Dear authors,

I suggest the following to improve the manuscript:

  1. Have the manuscript revised by a native English speaker familiar with the subject.

Response: the paper has been revides by a native English as requested by the reviewer

2. Did you plan on doing a systematic review? If yes, please report if you have registered it or not? Please report it in the form of a systematic review. If not, why didn't you do it as a systematic review?

Response: it's a systematic review.  Unfortunately we did not register in Prospero, and we can't register now because the research has already been completed and the system accepts only proposals of review. 

3. In your exclusion criteria, you state that studies were excluded if they did not report at least 5 years of follow-up, but in your tables and results you also talk about studies with follow up of less than 5 years. Please explain.

Response: We have eliminated from the table's works with less than 5 years of f-up.  They were not included in the count.

4. In line 264 you state "MV Z-score (OR=2.2, P=0.002, with an odds ratio of 1 for 1 unit decrease in Z-score).", what do you mean by odds ratio of 1 for 1 unit decrease?

Response: we have eliminated this confusing sentence

5. Describe what the AoV and mitral valve z score is.

Response: we have detailed z-scores sources and discussed within the text

Comments on the Quality of English Language

The English language needs improvement.

Response: the paper has been revides by a native English as requested by the reviewer

Reviewer 2 Report

Comments and Suggestions for Authors

I reviewed the manuscript, “Manuscript ID: healthcare-2755965. Type: Review, Title: N Neonates and Infants with Left Heart Obstruction and Border- 2 line Left Ventricle Undergoing Biventricular Repair: What We Know on Long Term Outcome? A Critical Review.” with great pleasure.

This study critically assesses the prognosis of neonates and infants with left ventricle outflow obstructions (LVOTO) and a borderline left ventricle (BLV) undergoing biventricular repair (BVR). A systematic search in February 2023 yielded 15 relevant studies from an initial pool of 51. These studies, using heterogeneous definitions for BLV, fell into three categories: those evaluating BLV, those focused on Shone’s complex, and those concerning aortic stenosis. Despite the variation in cardiac defects and eras, the results suggest a favorable survival rate and clinical outcome post-BVR, albeit with a high rate of reintervention and concerns about residual pulmonary hypertension, which remains under-explored. The current data on long-term outcomes for these patients is insufficient and fragmented, highlighting the need for larger, more comprehensive studies to understand these complex defects fully. This review constitutes a highly valuable contribution, encapsulating the diagnosis and prognosis of left ventricular stenosis lesions, a category of congenital heart disease, spanning from the neonatal period through to childhood. I would like to offer several recommendations for consideration.

1.      Concerning Table 1, merely cataloging data from previous publications does not constitute an adequate review. Specifically, in Table 1, regarding the definition of borderline left ventricle, it would be beneficial to include detailed information, such as the number of criteria used, whether the mitral valve and aortic valve annulus diameters are assessed using z-scores or actual measurements, and the specific types of congenital heart malformations involved. A diagram or figure summarizing these classifications and criteria would greatly enhance the utility and clarity of the information presented.

2.      Similarly, Table 2 requires refinement as its current form, merely listing results from previous studies, does not provide a meaningful summary. As highlighted in the discussion, the average prognosis of biventricular treatment for borderline left ventricle in the selected studies likely varies with age. Thus, it would be advantageous to collate and present this data, including average prognosis and reintervention rates, in an age-stratified manner. A comprehensive figure summarizing these aspects across various reports would significantly enhance the table's utility and informativeness.

3.      The Discussion section appropriately addresses a range of issues pertinent to congenital heart disease characterized by left ventricular stenosis lesions.

Author Response

Many thanks for your revision

Comments and Suggestions for Authors

I reviewed the manuscript, “Manuscript ID: healthcare-2755965. Type: Review, Title: N Neonates and Infants with Left Heart Obstruction and Border- 2 line Left Ventricle Undergoing Biventricular Repair: What We Know on Long Term Outcome? A Critical Review.” with great pleasure.

This study critically assesses the prognosis of neonates and infants with left ventricle outflow obstructions (LVOTO) and a borderline left ventricle (BLV) undergoing biventricular repair (BVR). A systematic search in February 2023 yielded 15 relevant studies from an initial pool of 51. These studies, using heterogeneous definitions for BLV, fell into three categories: those evaluating BLV, those focused on Shone’s complex, and those concerning aortic stenosis. Despite the variation in cardiac defects and eras, the results suggest a favorable survival rate and clinical outcome post-BVR, albeit with a high rate of reintervention and concerns about residual pulmonary hypertension, which remains under-explored. The current data on long-term outcomes for these patients is insufficient and fragmented, highlighting the need for larger, more comprehensive studies to understand these complex defects fully. This review constitutes a highly valuable contribution, encapsulating the diagnosis and prognosis of left ventricular stenosis lesions, a category of congenital heart disease, spanning from the neonatal period through to childhood. I would like to offer several recommendations for consideration.

Response: We thank the reviewer for the appreciation of our work

  1. Concerning Table 1, merely cataloging data from previous publications does not constitute an adequate review. Specifically, in Table 1, regarding the definition of borderline left ventricle, it would be beneficial to include detailed information, such as the number of criteria used, whether the mitral valve and aortic valve annulus diameters are assessed using z-scores or actual measurements, and the specific types of congenital heart malformations involved. A diagram or figure summarizing these classifications and criteria would greatly enhance the utility and clarity of the information presented.

Response: we have tried to detail better data on specific CHD and z-scores when available. For clarity of the reader we have dived the table according to the main types of criterion employed to define borderline LV (2 or more diminutive left sided anomalies , LVOT critical obstruction, Aortic coarctation + diminutive left sections)

  1. Similarly, Table 2 requires refinement as its current form, merely listing results from previous studies, does not provide a meaningful summary. As highlighted in the discussion, the average prognosis of biventricular treatment for borderline left ventricle in the selected studies likely varies with age. Thus, it would be advantageous to collate and present this data, including average prognosis and reintervention rates, in an age-stratified manner. A comprehensive figure summarizing these aspects across various reports would significantly enhance the table's utility and informativeness.

Response: we have tried to present data in an age stratified manner as suggested by the reviewer

  1. The Discussion section appropriately addresses a range of issues pertinent to congenital heart disease characterized by left ventricular stenosis lesions.

Response: We thank the reviewer for the appreciation of our work

Round 2

Reviewer 1 Report

Comments and Suggestions for Authors

Dear authors, 

Hi, 

I have several suggestions regarding my previous comments: 

1. You state in your rebuttal that this is a systematic review. If so, please state it also in the manuscript and report it like a systematic review based on the PRISMA guidelines. 

2. Use PRISMA flowchart for showing the flow of the study selection. 

3. Why did you only choose to search one database? In systematic reviews, it is best to search at least 3 databases. 

4. Include your complete search strategy and terms. 

5. I asked why did you include studies with less than 5 years of followup, and you said you have removed them, but I see that you have changed your inclusion to more than 3 years followup. Could you explain? 

Comments on the Quality of English Language

-

Author Response

Comments and Suggestions for Authors

Dear authors,

Hi,

I have several suggestions regarding my previous comments:

  1. You state in your rebuttal that this is a systematic review. If so, please state it also in the manuscript and report it like a systematic review based on the PRISMA guidelines.

Response: we have specified in the text

  1. Use PRISMA flowchart for showing the flow of the study selection.

Response: we have included a PRSIMA flow chart

  1. Why did you only choose to search one database? In systematic reviews, it is best to search at least 3 databases.

Response: we did search in 3 different database with similar results. We have explained that.

  1. Include your complete search strategy and terms.

Response: we have dome it

  1. I asked why did you include studies with less than 5 years of followup, and you said you have removed them, but I see that you have changed your inclusion to more than 3 years followup. Could you explain?

Response: we have removed studies with very limited follow-up, but we prefer to leave those studies with at least 3-years follow-up to avoid losing relevant information. We also recked in all the previously excluded papers, and there we no more paper to include extending the inclusion criteria to a follow-up of 3 years (instead of the 5 years initially defined).